# Trends Assessing Neuromuscular Fatigue in Team Sports: A Narrative Review

**DOI:** 10.3390/sports10030033

**Published:** 2022-02-28

**Authors:** Claudia Alba-Jiménez, Daniel Moreno-Doutres, Javier Peña

**Affiliations:** 1Sport and Physical Activity Studies Center (CEEAF), University of Vic, Central University of Catalonia, 08500 Barcelona, Spain; claudia.alba@uvic.cat (C.A.-J.); javier.pena@uvic.cat (J.P.); 2Sport Performance Analysis Research Group (SPARG), University of Vic, Central University of Catalonia, 08500 Barcelona, Spain; 3Club Joventut Badalona, 08912 Barcelona, Spain

**Keywords:** performance, monitoring, testing, objective measures, subjective measures

## Abstract

Neuromuscular fatigue is defined as a reduction induced by exercise in the maximal voluntary force that a muscle or group of muscles can generate. An accumulation of work or an incomplete force restoration can significantly influence the neuromuscular performance in both the short and long terms. Thus, fatigue management is essential for controlling the training adaptations of athletes and reducing their susceptibility to injury and illness. The main individualized monitoring tools used to describe fatigue are questionnaires and subjective assessments of fatigue, biochemical markers, sprint tests, and vertical jump tests. Among the subjective measures, the rating of the perceived exertion has been widely used because of its simplicity and high validity. In terms of the objective measures, one of the more frequently employed tools by practitioners to assess neuromuscular fatigue is the countermovement jump. Because of its high validity and reliability, it is accepted as the reference standard test in sports, in general, and particularly in team sports. Our review aims to clarify how all these indicators, as well as several devices, can help coaches in different sports contexts to monitor neuromuscular fatigue, and how these procedures should be used to obtain data that can be used to make decisions in complex environments.

## 1. Introduction

The word “fatigue”, which comes from the Latin word, “fatigare”, has an original meaning of “to cause to break down”, or “to tire” [1]. Different disciplines have historically analyzed fatigue, and its meaning changes to best suit diverse fields of knowledge. In applied sports sciences, fatigue is described merely as the reduced capacity to obtain the desired performance output, which limits the physical and cognitive functions by the interactions between the fatigability in the performance and the perceived fatigability [2,3,4]. According to Enoka and Duchateau [5], homeostasis maintenance and the athlete’s subjective psychological state are the main factors that are related to perceived fatigability. By contrast, the contractile function and muscle activation seem to be the most relevant factors for performance fatigability. Short-term fatigue has a metabolic origin, while prolonged fatigue originates at the neuromuscular level. Both are important to ensuring sports performances [6].

Neuromuscular fatigue (NMF) is a reduction in the maximal voluntary force induced by exercise, with neuromuscular function changes that are due to repeated or sustained muscular contraction, and that are produced either at the peripheral or central levels, and that can be detected for upwards of 48 h to an extended period [7,8,9]. Peripheral fatigue is developed earlier at the neuromuscular junction, and then at a muscular level, and it may play the most relevant role in short-term muscular performance decreases. Central fatigue appears via voluntary muscle neural activation and tends to occur later. It may cause limitations when peripheral fatigue increases, acting as a potential mechanism to safeguard from further damage or injury [10,11]. Additionally, the development of NMF may be task-dependent, which explains why using task-specific conditions can be more helpful to understanding the evolution of fatigue in response to exercise demands that are repetitive [12].

The accumulation of fatigue or incomplete recovery can significantly influence the team sports performance, especially during regular competition with a congested fixture calendar, which can have acute and chronic harmful effects (Figure 1). If the fatigue sustained by players and their recoveries are not managed correctly, athletes can potentially be placed at a higher risk of impaired performance, or at a more significant risk of injuries [13,14]. Although NMF control is necessary, the time needed to recover the neuromuscular function fully is not well established. In the long term, it has been reported that 24–48 h of recovery are necessary to return the measures of the sprint and vertical jumps to their neuromuscular function baselines. Other research shows that the vertical jump performance is reduced post-match, and that recovery requires at least 72 h [9,15,16]. This decrement in function can last for up to four days after a demanding competition [13]. Furthermore, the results imply that different individuals show relevant differences in their recovery profiles because the recovery time after a stimulus can have an individual component [17]. Therefore, personalized recovery strategies in sports are needed because some athletes benefit more from using recovery strategies than others to restore their physiological values [18]. Psychological factors also seem to play a pivotal role in recovery in the enhancement of the subsequent performance in actions such as sprints [19]. Overall, these studies reinforce the importance of individualized monitoring. To illustrate the use of tests to understand the NM status, we want to highlight the work from Jimenez-Reyes et al. [20], which used a vertical jump test to individualize training doses.

After fatiguing exercise, the time course and short-term recovery mechanisms are largely dependent on the properties of the previous exercise bout and the recovery time [21]. Balsom et al. [22,23] investigated the relationship between different durations of successive bouts of work, different between-set recovery times, and fatigue in two scientific works. First, they modified the working time (15, 30, and 40 m) while maintaining the recovery time between bouts. The authors concluded that a 30-s resting period is enough to recover from the 15-m repetitions, while significant performance reductions were detected in the other two distances. Second, they modified the resting time (30, 60, and 120 s) using a fixed-distance sprint (40 m), and they found that 30 s was insufficient to maintain the performance, while 60 and 120 s allowed the athletes to maintain acceleration and limit the drop in the performance in the last 10 m. Hence, aerobic metabolism plays an essential role after high-intensity intermittent training by restoring homeostasis during short-term recovery periods, which minimizes the drop in the neuromuscular performance [24].

The purpose of the review is to describe the information available about the effect of neuromuscular fatigue on the sports performance, decreasing the motor control, and, consequently, the sports injuries. The existing methods to evaluate this marker and assess fatigue in high-performance contexts are proposed for the control of the training load and a better recovery.

**Figure 1 sports-10-00033-f001:**
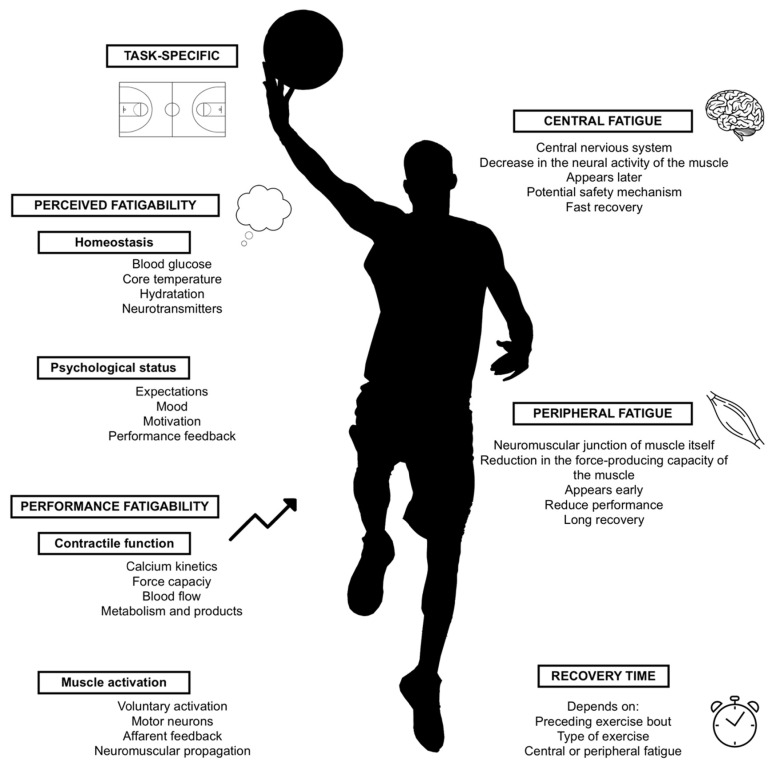
Sources and types of fatigue.

### 1.1. Tools to Monitor Neuromuscular Fatigue

The management of fatigue is essential for controlling the athletes’ training adaptations, for ensuring that they are ready for competition, and for reducing their susceptibility to injury and illness. In team sports, a reliable tool to monitor fatigue should be sensitive to the training loads and their magnitudes, and should differentiate the acute responses to exercise from the chronic changes [25].

#### 1.1.1. Athlete Self-Report Measures: Questionnaires and Subjective Assessments of Fatigue

The psychobiological state by prolonged periods of demanding cognitive activity (or mental fatigue) affects the individual perception of fatigue [26]. Mental fatigue drives athletes to downregulate their exercise capacity, which is known to be the maximum amount of physical exertion that an athlete can sustain [8]. Therefore, measuring these subjective markers is necessary to better understand NMF and recovery [27]. A recent survey on the use of fatigue-monitoring tools on high performance athletes in team-sport settings describes a high acceptance of the self-report questionnaires in various disciplines and competition levels to assess overall well-being [28]. The validated self-report forms are custom-designed forms, such as the Profile of Mood States Questionnaire (POMS), or the Recovery-Stress Questionnaire for Athletes (REST-Q), which are among the most widely used, and which may assist staff in monitoring the complex psychophysiological stress associated with high degrees of fatigue and poor recoveries [27]. The most regularly used is the rating of perceived exertion (RPE). The RPE is derived from a psychophysical process combining multiple sensations and feelings of physical stress, discomfort, and fatigue during exercise or physical activity [29]. Impellizzeri et al. [30] correlated the RPE with various methods to determine the internal training load, and they observe that it is a good indicator for it. This method may assist in the development of specific periodization strategies for individuals and teams. However, something relevant is that when questionnaires are implemented daily, their length should be considered. Many team sports practitioners prefer shorter and simpler questionnaires to minimize time constraints, which is more time-efficient when they have to be completed daily [4,27,31]. Implementing daily wellness questionnaires into an athlete monitoring program, such as the PAR-Q, requires time, but the RPE is a quick way to know the NM statuses of the athletes. A current study shows that a customized wellness questionnaire that encompasses the sleep quality, fatigue, muscle soreness, and mood on a 1–5 Likert scale produced an acceptable interday reliability, with a coefficient of variation (CV) of 6.9% [4]. Against this, some coaches raise concerns about the subjectivity and individual dimensions of these measures, as well as the scope for athletes to manipulate the responses to facilitate favorable outcomes [27]. Brito, Hertzog, and Nassis, in an article published in 2016 that assesses how the contextual variables influenced the training loads of highly trained soccer players under the age of 19, and they identified that the fatigue scores were inaccurate when using the sessional RPE (sRPE), and detected meaningful differences during the season. The individual fatigue scores that were reported varied significantly inside the microcycles [32]. The explanation for these inaccuracies may come from the fact that the perception of effort is very multidimensional and is determined by physiological, psychological, and experiential factors, as was determined by Morgan in a classic piece of research on the psychological components of the effort sense that was published in 1994 [33]. Moreover, the assessment of fatigue can be provided by the coach [3]. The performance markers can assist the coaching staff when an athlete is in a state of fatigue or recovery. There are available a multitude of fatigue markers to inform the coaching staff, and while the research in this area is plentiful, no single reliable diagnostic marker has been identified.

#### 1.1.2. Biochemical Markers

The acute responses and recovery times after practices and competitions can be assessed using diverse biochemical, hormonal, and immunological markers that are measured from the blood or saliva [27]. The endocrine system plays a relevant role when periodizing the workloads of athletes, which involves optimizing the training adaptations and avoiding further fatigue [34].

The most used biochemical markers to evaluate the responses to different workloads, training stresses, and recoveries are testosterone and cortisol [4]. Testosterone is an anabolic hormone, which plays a critical role in muscle hypertrophy and muscle glycogen synthesis [35], and it is also a neural facilitator that could affect the motor unit excitability [36]. Cortisol is a stress hormone, and is an indicator of the endocrine system’s response to exercise [34]. The independent responses to cortisol and testosterone have been widely studied, along with the changes in the anabolic–catabolic balance, or the testosterone:cortisol ratio (T:C), which are often observed [3]. The T:C ratio represents the imbalance between the anabolic and catabolic states, or the response to the training load, and it has been widely employed as a physiological signal to determine the anabolic and catabolic activity during increased workload periods [37]. It is hypothesized that an increase in the training load will decrease the T:C ratio, which shows an imbalance in the anabolic and catabolic responses [38]. A lowered T:C ratio means that players endure a catabolic hormonal profile for up to 24 h after a game. Thus, the relationship between testosterone and cortisol has been used as evidence of increased anabolic and catabolic activity during periods with high training loads, with the data indicating that decreases of 30% or over are the relevant markers of overreaching [37]. Creatine kinase (CK) is another relevant fatigue marker in athletes and players [3,4]. The CK enzyme is stored inside the muscle cells, but it is often released into the bloodstream after heavy exercise, which indicates muscle damage. Although the evidence appears compelling for CK’s use as a fatigue-monitoring tool in team sports, large individual variability in the resting CK levels exists, which makes it problematic to measure the changes induced by training [4]. CK has also shown large individual variability, with a high day-to-day variation of nearly 27% [27]. After all, several measures display low reliability and substantial intraindividual differences, which makes it challenging to obtain accurate measurements [3]. Moreover, the time, cost, and expertise required for the data collection and analysis are all high, the analysis is time-consuming, and there is generally a relatively long lag time to obtain feedback. These methodological limitations limit their use in high-performance environments and potentially impair the usefulness of such markers in a cyclic fatigue-monitoring system. The precise control of the previous exercise, the time of the day, the diet, the presence of injuries, the inconvenience of taking venipuncture blood samples, the possible unwillingness of some players to be subjected to invasive tests, and the relatively high cost associated with laboratory analysis, make this method difficult to implement in a practical training environment [3]. Moreover, the temporal relationships to the neuromuscular performance are not well established yet, and the multifaceted nature of fatigue makes it difficult to rely on a single biochemical, hormonal, or immunological marker [27].

#### 1.1.3. Surface Electromyography

Electromyography (EMG) refers to the collective electric signal from the muscles controlled by the nervous system that is produced during muscle contraction [39]. The EMG signal results from many physiological, anatomical, and technical factors. Proper detection methods may manage the effects of some of these factors, but others are not easily regulated with the current technology, and their potential effects on the signals may only be surmised and considered [40]. There are two types of EMG signals: surface EMG and intramuscular EMG. Surface EMG (sEMG) is preferable when obtaining information about the time or intensity of the superficial muscle activation with noninvasive electrodes [39]. The sEMG signal is used as an indicator of the muscle activation for its relationship to the force produced by a muscle, and as an index of the fatigue processes occurring within a muscle [40]. Thus, sEMG signals are related to the skeletal musculature’s biochemical and physiological changes during fatiguing contractions [41]. It is also applicable to the study of static actions that require a muscular effort of a postural type, but its use is limited to those involving dynamic movements. Dynamic actions have to be synchronized with the recording of the other measurement systems that provide the cinematic data as a camera [42]. Its principal advantages are its noninvasiveness, its applicability in situ, the real-time fatigue monitoring during the performance of the defined work, the ability to monitor the fatigue of a particular muscle, and the correlations with the biochemical and physiological muscle changes during the fatiguing [41]. It is evident that sEMG has several advantages, but it has severe reliability problems, and it is still challenging to validate the relationships observed between the sEMG parameters and the physiological events. The lack of standards for the sensors, configurations, electrode placement, and recording protocols has adversely affected the possibility of its integration into the team sport context [43].

#### 1.1.4. Sprinting Ability

Sprint tests have been widely used to describe the NMF of athletes and their performances in various sports, and most of them use similar sprint distances, such as 5, 10, and 20 m [4,17,27,44,45,46,47]. Very large correlations have been found between the speed loss and the lactate (r = 0.83) and ammonia (r = 0.86) concentrations when the metabolic responses to the sprint training are focused on maintaining a maximal speed until a given speed loss is reached [20]. These findings support the use of sprint tests as an excellent tool for determining the fatigue responses because of their agreement with the physiological gold-standard measures. Different sprint distances have been studied to improve our understanding of how NMF and performance interact. Thus, the type of sprinting (running, rowing, skiing, or leg or arm cycling), the number of sprints, the length of each sprint, the time to recover between sprints, and the work-to-rest ratios of a sprint have been analyzed. These factors may vary the sprint performance, thus affecting the NMF interpretation. Using task-specific parameters may help to understand the development of NMF when responding to a repeated sprint exercise in a given sport [12]. Hence, many authors agree that this drill is the most task-specific measure of NMF [27,44,46,47]. According to Marrier et al. [46], in team sports, such as rugby sevens, sprint accelerations and decelerations are more frequent than vertical jumps (VJ). Sprint tests rely primarily on the concentric muscle action, whereas the VJ fundamentally relies on a stretch-shortening cycle (SSC) [47]. A running test could be more sensitive to neuromuscular status changes than a jump test because of the higher task-specific nature. Garret et al. [44] observed this trend in Australian Football, a predominantly running sport. Surprisingly, the authors found similar results between the sprint test and the vertical jump tests, which shows an alternative method of assessing the neuromuscular function in high-performance athletes. In basketball, the sprint speed has been identified as a relevant attribute; specifically, 5 m of sprint time showed a moderate inverse relationship to the playing time in the NCAA Division II competition. Thus, monitoring the athletes’ acceleration abilities can be a more suitable method of identifying fatigue, in opposition to the maximal speed. The sprint performance could be considered a valid tool for the assessment of NMF in sports where sprints are specific to the task [20]. Since it does not cause excessive fatigue, it is easy to administer as a part of the warm-up, and it can be applied to large groups of athletes concurrently in a different number of environments and settings, i.e., indoors and outdoors, which increases its ecological validity [44]. However, some authors report that the sprint performance was less sensitive as a fatigue marker compared with the CMJ, which suggests that its use to profile recovery is limited [17].

#### 1.1.5. Vertical Jump Tests

The benefit of vertical jumps as a practical measure of NMF is indicated by the high degree of the adoption of these testing procedures in high-performance sports settings. These tests have been used in many studies to investigate the recovery times from demanding practices or competitions [28]. Vertical jump tests are practical, well accepted by elite players, and are valid and reliable, which makes them potentially valuable for detecting and quantifying fatigue in in-field conditions [46]. Furthermore, jump tests allow prolonged superior sensitivity to altered neuromuscular function, and likely, NMF reflects the stretch-shortening capability of the lower-limb muscles and the ability to assess muscular fatigue [4,27]. Jump tests are quick and easy to implement, with many of the techniques scientifically validated. Moreover, there are reliable technologies available for adopting them that cause minimal additional fatigue. In sports such as basketball or volleyball, athletes perform many offensive and defensive movements while training and competing, including accelerations, decelerations, and COD, which rely on the athlete’s ability to transition from eccentric to concentric contractions, which is effectively the SSC [48,49]. The repeated execution of these movements can result in diminished movement efficiency through NMF and performance fatigue [4]. Additionally, vertical jumps remain stable during and across multiple days. This stability may be due to the athletes regularly performing multiple jumps through training and competition, which results in more reproducible movement patterns. The variables related to the vertical jump output and the loading strategy exhibit acceptable trial-to-trial and interday reliabilities, although some jumps are more reliable than others. In recent research, the authors of [2] suggest that the mean force, the mean power, and the relative mean power should be used by practitioners, as they exhibit both acceptable reliabilities and sensitivities. Conversely, Taylor et al. [28] also asked strength and conditioning coaches and sport science professionals about vertical jump tests and jump assessment protocols. The respondents indicated that the jump height is still the most popular variable assessed in fatigue-monitoring systems. However, numerous other kinetic and kinematic variables, such as the peak and mean velocities, the peak and mean powers, and the peak force, were also monitored. Apart from the jump height, the variable flight time to the contraction time (FT:CT) of the vertical jump test is a valid tool for assessing NMF [2,50].

Several Vertical Jump Tests Exist [51]. Nevertheless, the squat jump (SJ), the countermovement jump (CMJ), and the drop jump (DJ) are the usual jump tests that are used within the literature [27,28,45,52]. Even though the use of the three jumps mentioned above to monitor fatigue is well documented, the CMJ is the most popular vertical jump test among practitioners for assessing NMF [4,15,16,17,44,45,52,53]. Taylor et al. [28] studied the current trends through a questionnaire, and for the performance tests, the VJ was the most popular and was used by 54% of all the responders. The VJ test is the performance test that produces less fatigue when compared with the sprinting or strength tests [27]. While the simple measures of the jump performance are cheap and easy to administer with large groups, they are helpful because, as described above, they reflect the stretch-shortening capability of the lower-limb musculature and the ability to evaluate muscle fatigue [27].

### 1.2. Countermovement Jump

The countermovement jump (CMJ) is the vertical jump test that is more frequently used to assess the jumping performance and the neuromuscular status. Previous works have studied its validity and reliability compared to other VJs (Table 1).

Many factors can influence the CMJ. The main factors cited in the bibliography are:

The countermovement depth: The protocols describe that the jump initiates with the participants in an upright position before executing the vertical jump, which starts with a countermovement until the legs are bent down to 90° [58]. Despite this, a protocol wherein a self-selected knee angle is used may present higher reliability, and a short test duration should potentially minimize errors [37,56,57,59,60,61,62,63,64,65];

The arm swing: The arm swing influences the vertical jump performance, and increases the jump height compared to vertical jumps without an arm swing [54,57,60,63,66,67]. Despite the performance improvement, the lower variability due to less error of measurement, a smaller average measurement bias, a reduction in the measurement difference variability, and a higher reliability of the CMJ without the arm swing versus the CMJ with the arm swing suggests that maintaining the arms in a fixed position provides a more stable form than allowing unrestricted arm movement [63,68].

The jumps considered for the analysis are another factor of the variability between the studies. A meta-analysis of the CMJ test to monitor the neuromuscular status determined a predominance of studies using the highest CMJ performance value for their analyses. However, when comparing the highest and average results, the average jump results were more sensitive than the highest jumps in identifying fatigue or the effects of supercompensation [69]. A systematic review of the CMJ and the SJ defined the most common numbers of trials performed in the published research and found that three jumps were found 76% of the time, compared to two jumps (11%), and more than three jumps (13%) [58].

Despite being the most commonly used test to evaluate the vertical performance, there is no general agreement in terms of the CMJ protocol. The most common description in the literature follows the upcoming rules: the CMJ has to be performed with the participants’ hands on their hips and starting from an upright static position with their legs straight. The participants have to be instructed to squat by bending the knees at approximately 90° angles as quickly as possible. Then, they should jump as high as possible, keeping the legs straight, and landing with both feet together [54,61,70,71,72,73,74]. In a recent study, McMahon et al. [75] offer a detailed description of the CMJ phases (Table 2).

### 1.3. Is Countermovement Jump a Valid Tool to Measure Neuromuscular Fatigue?

The countermovement jump (CMJ) is one of the main tools used to examine the level of the neuromuscular status in elite sports. Because of its reliability and validity, the CMJ test has become the “gold standard” test for monitoring neuromuscular fatigue in high-performance sports settings [44]. Other authors also suggest that its high repeatability and fatigue sensitivity prove its usefulness, and it is currently the most valid test for detecting neuromuscular fatigue (NMF) [45].

There have been many attempts to define the fatigue induced in the neuromuscular function through CMJ assessments in team-sport athletes. In a scientific work by Gathercole et al. [15], the intraday and interday reliability comparisons indicated high reliability, with an absence of systematic changes in the CMJ reproducibility. With an intraday CV of 5.3, and an interday CV of 4.9%, CMJ testing could be a proper noninvasive method to use in athlete NMF monitoring. The reliability of the CMJ to measure NMF has also been studied in other sports, such as soccer, rugby, and basketball. In semiprofessional soccer players, the CMJ has excellent test–retest reliability for measuring NMF [52]. The authors describe an ICC of 0.88 and a CV of 4.8% in the variable jump height. In Australian Rules Football, the results confirm the jump height as a sensitive measure of NMF after a match play, with a CV of 8.5% and a smallest worthwhile change of 1% [44]. In professional rugby league players, McLean et al. [13] used the CMJ to monitor NMF, and they suggest that regular analyses of the CMJ are valuable tools for monitoring in-season fatigue. Roe et al. [47] have demonstrated that the CMJ metrics are useful for monitoring the lower-body neuromuscular function in rugby union players. In basketball, the CMJ has also been studied as a tool to measure NMF. A study carried out with professional and semiprofessional basketball players found a high test–retest reliability, with an ICC of 0.82 and a CV of 3.8% in the jump height variable [76]. In another study on Division I men’s collegiate basketball, Edwards et al. [2] studied the reliability and sensitivity of NMF through the VJ. The jump height expressed acceptable trial-to-trial reliability, with a CV of 5.6%, and a smallest worthwhile change of 2.4%. However, this variable showed lower interday reliability, with a CV of 12.1%. The conflicting interday reliability only results from the jump height variable. For example, the variable’s peak force and mean force showed high interday reliabilities, with CVs of 3.1% and 3.8%, respectively. Apart from the jump height, the vast number of variables exhibiting acceptable reliability suggests that the CMJ strategy and the output remain stable during and across multiple days. This stability may be attributed to the basketball athletes regularly performing multiple jumps and SSC activities, such as COD through training and competition, which results in more reproducible movement patterns. Along the same idea, Spiteri et al. [16] also studied NMF in basketball and they indicate the the FT:CT appears to be a sensitive measure for monitoring the training intensity and for detecting NMF following training and game performances.

### 1.4. Technologies to Measure the Vertical Jump

There are several technologies available on the market to measure the CMJ:

Force platforms (FPs): Force platforms are considered the “gold standard technology”, which measure the force exerted on it by the subject [77];

Contact mats (CMs): These are electric circuits that are mechanically activated by pressure, and most of them use the flight time to indirectly estimate the jump height [78,79];

Photoelectric cell (PC) systems: These systems measure the flight times with two parallel bars: one receiver unit and one transmitter unit [80];

Local positioning systems (LPS): These are based on either global positioning systems (GPS) or indoor positioning systems, with accelerometers and gyroscopes integrated into the device to calculate and perform corrections to the vertical acceleration recordings [70];

Phone apps: These are based on the detection of the initial and final phases of the jump through high-speed recording technology [81];

Accelerometers: These classify movements in the vertical axis as jumps, and quantify the vertical component of each jump using a proprietary algorithm [7].

Traditionally, the validation of new technologies to assess jumps has been carried out using FPs. When FPs were compared with CMs, Steinman, Shirley, and Fuller [82] described similar jump heights with high ICCs; contrarily, Withmer et al. [83] found no consistent results in the flight time variables between both tools. Nonetheless, the CM is considered a valid technology for measuring the VJ performance [82,84]. For the PC, Attia et al. [71] determined a high correlation between PCs and FPs for the measurement of the vertical jump height, despite a present systematic bias, while Glatthorn et al. [80] found an excellent test–retest reliability between both technologies. For the LPS, a descriptive analysis shows that the flight time recorded by an inertial device was almost equal to the one registered by the FP, and there were no meaningful differences between the devices [70]. A recent review on the use of a phone app for jump-based diagnostics also employed the FP as a “gold standard” to assess the validity and reliability of that technology [81,85,86]. Accelerometers have also been validated to FPs, with the results describing an overestimation in the jump heights [68,87].

The main advantages and disadvantages of the presented technologies are described in Table 3.

## 2. Training Program and Fatigue

Monitoring team sports activity and its recovery can inform athletes’ fatigue [16]. Furthermore, there is also a need to ensure the appropriate monitoring of individuals within a team environment. Athletes may respond differently to the training stimulus, and the training loads required for the adaptation may differ significantly, and consequently, so may the fatigue that is produced by the training load. Monitoring the individual athlete allows for the identification of those athletes who are not responding to the training program, and for the control of the internal and external loads to avoid the appearance of fatigue [96]. Moreover, the movement technique, or the agility, are related to the performance and influence fatigue [48]. When local muscular work is relatively heavy, and of a considerable duration, the fatigue it causes is transferred to and impairs both the speed and the accuracy in the neuromotor-coordination tasks performed by these and the associated muscles [97]. There is extensive literature about team sports and NMF. The performance tests have been validated with high reliability in professional soccer, rugby, and basketball teams. However, there is no evidence of the different player positions in any sport [44,47,52].

## 3. Conclusions

The present narrative review describes NMF and the complex processes that cause this specific type of fatigue. NMF has been reported as a reduction induced by exercise in the maximal voluntary force produced by a muscle or a group of muscles, and to understand its extent is pivotal because of its consequences on sports performances and athlete statuses. Various NMF monitoring procedures have been used in the past, but not all of them are suitable in team sports. For example, in team sports, biochemical markers are not used for their methodological limitations. The questionnaires and subjective assessments of fatigue are not accurate because the perception of effort and fatigue is multidimensional. However, performance tests (sprinting ability and vertical jump test) are the most used for their practical application in team sports training. Furthermore, not all these monitoring strategies provide the same information on how athletes respond to training and nontraining stressors. This article reviews the main technologies used and their advantages and disadvantages in terms of the cost, the time needed to gather and process the information obtained, as well as in terms of the validity and reliability. We recommend that coaches and practitioners decide which are the most appropriate for their particular situations, but ecology is the most important of these procedures in high-performance sports settings.

## Figures and Tables

**Table 1 sports-10-00033-t001:** Validity and reliability of the CMJ in different studies.

Study	Participants		ICC	CV%	Cronbach
Markovic et al. [51]	*n* = 93, health collegiate athletes		0.98	2.8	0.98
Slinde et al. [54]	*n* = 30, recreational athletes		0.93		0.96
Richter et al. [55]	*n* = 324, secondary school athletes		0.96	4.4	
Gathercole et al. [15]	*n* = 11, college-level team-sport athletes	Intersession		4.9	
		Intrasession		5.3	
Byrne et al. [56]	*n* = 18, hurling players		0.95	5.5	0.95
Heishman et al. [57]	*n* = 22, NCAA Division 1 collegiate basketball players	Intersession	0.96	4.7	0.99
		Intrasession	0.97	3.8	0.99
Fitzpatrick et al. [52]	*n* = 17, elite youth soccer players		0.88	4.8	

ICC = interclass correlation; CV% = coefficient of variation; Cronbach = Cronbach’s α.

**Table 2 sports-10-00033-t002:** CMJ phases. Adapted from McMahon et al. [75].

Weighting Phase	Athlete is Required to Stand as Still as Possible for 1 s.
Unweighting phase	Athlete starts the countermovement with a combined flexion of the hip, knees, and ankles.
Braking phase	Athlete decelerates their center of mass (COM), which coincides with the deepest part of the squat. The leg extensor muscle-tendon units are actively stretching to decelerate the body mass.
Propulsion phase	Athlete extends hips, knees, and ankles. Positive COM velocity is achieved.
Flight phase	Athlete leaves the floor (it starts at the take-off and ends at the touchdown). Maximal positive COM displacement.
Landing phase	COM velocity deceleration Athlete “absorbs” the landing by flexing the ankles, knees, and hips.

**Table 3 sports-10-00033-t003:** Advantages (pros) and disadvantages (cons) of the presented technologies.

Technologies	Pros	Cons	Suitability in Team Sports	
Force platforms	- High levels of precision and accuracy in VJ test [88] - Kinetic and temporal variables producing force–time curves [78]	- Expensive [88] - Bulky [88] - Hard to transport [88] - Specific computer software [88] - Limited to a laboratory setting/No sport-specific usability [89]	- Price - Measurement time - Processing time - Reliability and validity - Ecology	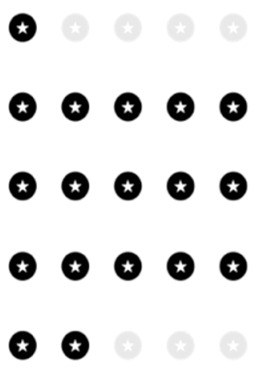
Contact mats	- Easy to transport [90] - Low cost [90] - High accessibility [90]	- Specific computer software - Feet are not directly in contact with the specific sport surface [80] - Indirectly measures the jump height by flight time [91]	- Price - Measurement time - Processing time - Reliability and validity - Ecology	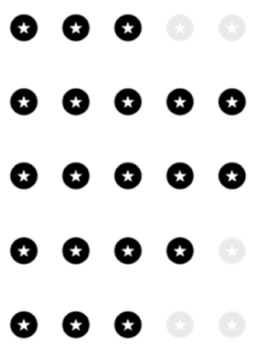
Photoelectric cells system	- Easy to transport and easy to handle [80] - Can be placed on all sports surfaces [91] - Relatively cost-effective compared with FPs [80]	- Specific computer software - Indirectly measures the jump height by flight time [91] - Expensive compared to CMs [81]	- Price - Measurement time - Processing time - Reliability and validity - Ecology	
Local positioning systems	- Lightweight and portable [70] - Easily placed in any segment of the body [70] - Data from many subjects can be obtained at the same time [70] - Subjects do not have to be connected by any cable, nor need to take-off or land in a delimited area [70] - Immediate feedback [70]	- Expensive [70] - Lack of validity of some devices [92] - Lack of transparency of some companies in reporting device validity [92] - Differences resulting from different software versions from the same company [92] - Differences between companies with the same types of technologies [92]	- Price - Measurement time - Processing time - Reliability and validity - Ecology	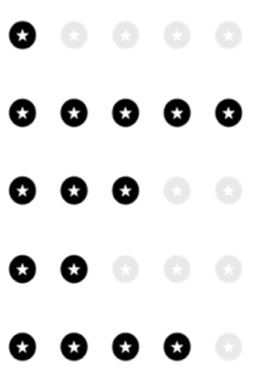
Phone apps	- Affordable [93] - Portable [81,93] - Inexpensive [81] - Frequently available on both Android and iOS [93]	- Absence of scientific works reporting their reliability [86] - A determination of the stability of phone apps in quantifying the jump performance over time is needed [86]	- Price - Measurement time - Processing time - Reliability and validity - Ecology	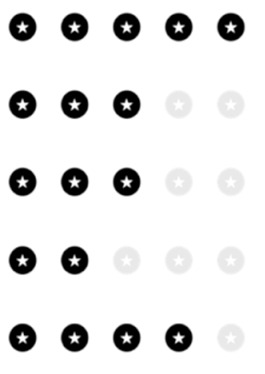
Accelerometers	- Simple [94] - Inexpensive [94] - Monitor athletes in real-time during training and official matches [95] - Provide real-time information in the moment to coaches [78]	- Algorithms are not available for public inspection [94] - Unclear thresholds [94]	- Price - Measurement time - Processing time - Reliability and validity - Ecology	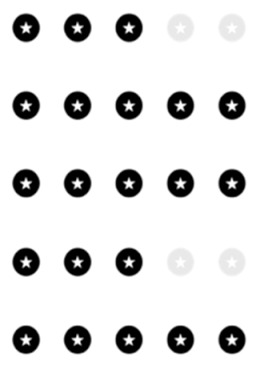

Suitability: from 0 (poor) to 5 (excellent).

## Data Availability

Not applicable.

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
