# Peer review of "Trends Assessing Neuromuscular Fatigue in Team Sports: A Narrative Review"

_sports, 2022, doi:10.3390/sports10030033_

Round 1
Reviewer 1 Report
First of all, I would like to praise the authors and thank them for choosing a topic that is of great importance for the field of high performance team sports.
This paper presented the relevant part of the current knowledge and trends about the tools used to assess the NMF in team sports.
In order to improve the paper, I suggest certain interventions:
- Assessment of athlete fatigue, and thus neuromuscular fatigue, can be provided by registering the subjective assessment of the coach (body language, quality and speed of specific movement, speed of decision making in specific situation, quickness and reactions…). Therefore, I would recommend the authors to at least open this topic in the text in the introduction or in the conclusion.
- The ultimate purpose of NMF assessment is about interventions in training program design. Therefore, I suggest that the authors mention this process/connection in a short chapter before concluding.
- The technique of performing the technical elements of a particular sport is highly related to the NMF. Athletes whose technique is more rational and economical feel fatigue later and less during specific training and competition. I suggest the authors to briefly look at this phenomenon in the context of the importance of motor learning in the training process.
- In the conclusions and practical application, it is important to look more extensively at personalized assessment for each athlete and at the specifics of the sport and players position specifics in the sport. Therefore, I suggest the authors to present their thoughts on this important topic and maybe to create separate table (not obligated) related to sport/position specific demands on NMF assessment.
Author Response
First of all, I would like to praise the authors and thank them for choosing a topic that is of great importance for the field of high-performance team sports.
This paper presented the relevant part of the current knowledge and trends about the tools used to assess the NMF in team sports.
In order to improve the paper, I suggest certain interventions:
R1: Assessment of athlete fatigue, and thus neuromuscular fatigue, can be provided by registering the subjective assessment of the coach (body language, quality and speed of specific movement, speed of decision making in specific situation, quickness and reactions…). Therefore, I would recommend the authors to at least open this topic in the text in the introduction or in the conclusion.
AR: Thanks for the comment. We have added this information in section 1.1.1. Athlete self-report measures, where we describe the subjective assessment of fatigue for the athletes.
“Moreover, the assessment of fatigue can be provided by the coach. The performance markers can assist the coaching staff when an athlete is in a state of fatigue or recovery. There are available a multitude of fatigue markers to inform the coaching staff, and while the research in this area is plentiful, no single, reliable diagnostic marker has been identified.
R1: The ultimate purpose of NMF assessment is about interventions in training program design. Therefore, I suggest that the authors mention this process/connection in a short chapter before concluding.
AR: Thank you for the comment. We have added this practical information in a new section.
“2. Training program and fatigue
Monitoring team sports activity and its recovery can provide information of athletes’ fatigue (Spiteri et al., 2013). Further, there is also a need to ensure appropriate monitoring of individuals within a team environment. Athletes may respond differently to the training stimulus, and the training load required for the adaptation may differ significantly, and consequently, the fatigue produced by the training load. Monitoring the individual athlete allows the identification of those athletes who are not responding to the training program and control the internal and external load to avoid the appearance of fatigue (Halson, 2014)“.
Halson, S. L. (2014, November 1). Monitoring Training Load to Understand Fatigue in Athletes. Sports Medicine. Springer International Publishing. https://doi.org/10.1007/s40279-014-0253-z
Spiteri, T., Nimphius, S., Wolski, A., & Bird, S. (2013). Monitoring neuromuscular fatigue in female basketball players across training and game performance. Journal of Australian Strength and Conditioning, 21(S2), 73–74.
R1: The technique of performing the technical elements of a particular sport is highly related to the NMF. Athletes whose technique is more rational and economical feel fatigue later and less during specific training and competition. I suggest the authors to briefly look at this phenomenon in the context of the importance of motor learning in the training process.
AR: Thanks for the comment. We have added this information in section 2. Training program and fatigue.
"Moreover, movement technique or agility are related to performance, influencing fatigue (Padulo et al., 2016). When local muscular work is relatively heavy, and of considerable duration, the fatigue it causes is transferred to and impairs both speed and accuracy in neuromotor-coordination tasks performed by these and associated muscles (Alderman, 1965)".
Alderman, R. B. (1965). Influence of local fatigue on speed and accuracy in motor learning. Research Quarterly of the American Association for Health, Physical Education and Recreation, 36(2), 131–140. https://doi.org/10.1080/10671188.1965.10614670
Padulo, J., Bragazzi, N. L., Nikolaidis, P. T., Dello Iacono, A., Attene, G., Pizzolato, F., … Migliaccio, G. M. (2016). Repeated sprint ability in young basketball players: multi-direction vs. one-change of direction (part 1). Frontiers in Physiology, 7(133), 1–12. https://doi.org/10.3389/fphys.2016.00133
R1: In the conclusions and practical application, it is important to look more extensively at personalized assessment for each athlete and at the specifics of the sport and players position specifics in the sport. Therefore, I suggest the authors to present their thoughts on this important topic and maybe to create separate table (not obligated) related to sport/position specific demands on NMF assessment.
AR: Thanks for the comment. We have added this information in a paragraph in section 2. Training program and fatigue.
“There is extensive literature about team sports and NMF. The performance tests are validated with high reliability in professional soccer, rugby, or basketball teams. However, there is no evidence of the different player positions in any sport (Fitzpatrick et al., 2019; Garrett et al., 2019; Roe et al., 2016)“.
Fitzpatrick, J. F., Hicks, K. M., Russell, M., & Hayes, P. R. (2019). The reliability of potential fatigue-monitoring measures in elite youth soccer players. Journal of Strength and Conditioning Research, (12), 1–5. https://doi.org/10.1519/jsc.0000000000003317
Garrett, J., Graham, S. R., Eston, R. G., Burgess, D. J., Garrett, L. J., Jakeman, J., & Norton, K. (2019). A novel method of assessment for monitoring neuromuscular fatigue in Australian rules football players. International Journal of Sports Physiology and Performance, 14(5), 598–605. https://doi.org/10.1123/ijspp.2018-0253
Roe, G., Darrall-Jones, J., Till, K., Phibbs, P., Read, D., Weakley, J., & Jones, B. (2016). To jump or cycle? Monitoring neuromuscular function in rugby union players. International Journal of Sports Physiology and Performance, 12(5), 690–696.

Reviewer 2 Report
Presented study aimed to review trends assessing neuromuscular fatigue in team sports
I have several suggestions:
Introduction:
- Please state the purpose of the review At the end of the introduction
Materials and Methods:
- Please add the material and methods section and include the criteria for selecting the publication for review:
- which databases were used and for what period of time?
- what keywords were used for the selection of the literature?
- what were the criteria for exclusion from work?
Results and discussion:
- Chapters 1.1 to 1.4 should be included in the results and discussion chapter.
- The sEMG parameters should also be assessed for this work (e.g., https://pubmed.ncbi.nlm.nih.gov/29936808/; https://pubmed.ncbi.nlm.nih.gov/17113787/)
I hope my suggestions will help improve this paper.
Author Response
R2: Presented study aimed to review trends assessing neuromuscular fatigue in team sports, I have several suggestions:
Introduction:
- Please state the purpose of the review at the end of the introduction
AR: Thanks for the comment. The following information has been added at the end of the corresponding paragraph:
“The purpose of the review is to describe the information about the effect of neuromuscular fatigue on sports performance, decreasing motor control and consequently, the sports injuries. The existing methods to evaluate this marker and assess fatigue in a high-performance context are proposed to control training load and recovery better”.
R2: Materials and Methods:
- Please add the material and methods section and include the criteria for selecting the publication for review:
- which databases were used and for what period of time?
- what keywords were used for the selection of the literature?
- what were the criteria for exclusion from work?
AR: Thanks for the comment. We decided not to add a material and methods section with the search procedures because this article follows the structure of a narrative review and not a systematic review nor a review from the same family. For Rother (2007), a narrative review article describes and discusses the state of the science of a specific topic or theme from a theoretical and contextual point of view. Narrative reviews do not list the types of databases and methodological approaches used to conduct the review or the evaluation criteria for inclusion. Furthermore, the sections that the author describes for a narrative review are the introduction, development (using necessary sub-headings to divide and discuss the topic appropriately), discussion, and references. Ferrari (2015) also differences the systematic review articles from narrative review articles and states there is no consensus on the standard structure of a narrative review. The narrative review structure should respect the author's preferences and confirm that the methods section is not mandatory, unlike systematic reviews.
Rother, E. T. (2007) Systematic literature review x narrative review. Acta Paulista de Enfermagem, 20, v-vi.
Ferrari, R. (2015). Writing narrative style literature reviews. Medical Writing, 24(4), 230-235.
R2: Results and discussion:
- Chapters 1.1 to 1.4 should be included in the results and discussion chapter.
AR: Thanks for the comment. We have modified the conclusion of the article with the main ideas of sections 1.1 and 1.4:
"Various NMF monitoring procedures have been used in the past, but not all of them are suitable in team sports. For example, in team sports, biochemical markers are not used for their methodological limitations. Questionnaires and subjective assessments of fatigue are not accurate because the perception of effort and fatigue is multidimensional. However, performance tests (sprinting ability and vertical jump test) are the most used for their practical application in team sports training. Furthermore, not all these monitoring strategies provide the same information on how athletes respond to training and non-training stressors. This article reviews the main technologies used and their advantages and disadvantages as cost, the time needed to gather and process the information obtained, or the validity and reliability. We recommend that coaches and practitioners decide which are the most appropriate for their particular situations, but ecology is the most important of these procedures in high-performance sports settings".
R2:
- The sEMG parameters should also be assessed for this work (e.g., https://pubmed.ncbi.nlm.nih.gov/29936808/; https://pubmed.ncbi.nlm.nih.gov/17113787/)
AR: Thanks for the comment. We agreed that sEMG is an excellent tool to assess NMF, so we added a new section in the article called 1.1.3. Surface electromyography in section 1.1. Tools to monitor neuromuscular fatigue.
"1.1.3. Surface electromyography. Electromyography (EMG) refers to the collective electric signal from muscles controlled by the nervous system and produced during muscle contraction (Chowdhury et al., 2013). The EMG signal results from many physiological, anatomical, and technical factors. Proper detection methods may manage the effects of some of these factors, but others are not easily regulated with current technology, and their potential effect on signals may only be surmised and considered (De Luca, 1997). There are two types of EMG signals: surface EMG and intramuscular EMG. Surface EMG (sEMG) is preferably used to obtain information about the time or intensity of superficial muscle activation with non-invasive electrodes (Chowdhury et al., 2013). The sEMG signal is used as an indicator of muscle activation for its relationship to the force produced by a muscle and its use as an index of fatigue processes occurring within a muscle (De Luca, 1997). So, sEMG signals are related to skeletal musculature's biochemical and physiological changes during fatiguing contractions (Cifrek et al., 2009). It is also applicable to the study of static actions requiring a muscular effort of a postural type, but its use is limited to those involving a dynamic movement. Dynamic actions have to be synchronized with the recording of other measurement systems that provide cinematic data as a camera (Massó et al., 2010). Its principal advantages are non-invasiveness, applicability in situ, real-time fatigue monitoring during the performance of defined work, the ability to monitor fatigue of a particular muscle, and correlation with biochemical and physiological muscle changes during fatiguing (Cifrek et al., 2009). It is evident that sEMG has several advantages, but it has severe reliability problems, and it is still challenging to validate the relationships observed between the sEMG parameters and physiological events. The lack of standards for sensors, configurations, electrode placement, and recording protocols has adversely affected the possibility of its integration into a team sport context (Hogrel, 2005).
Chowdhury, R. H., Reaz, M. B. I., Bin Mohd Ali, M. A., Bakar, A. A. A., Chellappan, K., & Chang, T. G. (2013). Surface electromyography signal processing and classification techniques. Sensors (Switzerland), 13(9), 12431–12466. https://doi.org/10.3390/s130912431
Cifrek, M., Medved, V., Tonković, S., & Ostojić, S. (2009). Surface EMG based muscle fatigue evaluation in biomechanics. Clinical Biomechanics, 24(4), 327–340. https://doi.org/10.1016/j.clinbiomech.2009.01.010
De Luca, C. J. (1997). The use of surface electromyography in biomechanics. Journal of Applied Biomechanics, 13(2), 135–163. https://doi.org/10.1123/jab.13.2.135
Hogrel, J. Y. (2005). Clinical applications of surface electromyography in neuromuscular disorders. Neurophysiologie Clinique, 35(2–3), 59–71. https://doi.org/10.1016/j.neucli.2005.03.001
Massó, N., Rey, F., Romero, D., Gual, G., Costa, L., & Germán, A. (2010). Surface electromyography applications. Apunts: Medicina de l’esport, 45(166), 121-131–131. https://doi.org/10.1016/j.apunts.2010.02.005

Round 2
Reviewer 2 Report
Congratulations to the Authors of the revised version of the article. Hope my comments improved the value of the work.